# Daily Rhythms in the IGF-1 System in the Liver of Goldfish and Their Synchronization to Light/Dark Cycle and Feeding Time

**DOI:** 10.3390/ani12233371

**Published:** 2022-11-30

**Authors:** Aitana Alonso-Gómez, Diego Madera, Ángel Luis Alonso-Gómez, Ana Isabel Valenciano, María Jesús Delgado

**Affiliations:** Department of Genetics, Physiology and Microbiology, Faculty of Biology, Complutense University of Madrid, 28040 Madrid, Spain

**Keywords:** insulin-like growth factor 1, insulin-like growth factor receptors, insulin-like binding proteins, circadian, gene expression, biological clock, chronobiology, fish

## Abstract

**Simple Summary:**

Insulin-like growth factor-1 is a peptide that acts both as a hormone and growth factor that participates in several physiological processes in vertebrates. Due to its pleiotropic actions, its functionality is regulated both by its linkage to binding proteins and the signaling through specific receptors. This system may be temporally synchronized to ensure the anticipatory physiological adaptations to cyclic environmental changes. The aim of this work is to study the daily variations in the expression of these components belonging to the insulin-like growth factor-1 system in the liver of goldfish, as well as the influence of two environmental synchronizers, the light/dark cycle and a scheduled feeding time, on such rhythmicity. Both environmental cues influence the daily rhythms of expression of the components of the insulin-like growth factor-1 system, and particularly, feeding time synchronizes these rhythms. Overall, this work evidences the relevance of insulin-like growth factor-1 and its binding proteins as potential rhythmic outputs of the liver clock in fish. A scheduled mealtime plays a key role in the functional organization of the circadian system in animals.

**Abstract:**

The relevance of the insulin-like growth factor-1 (IGF-1) system in several physiological processes is well-known in vertebrates, although little information about their temporal organization is available. This work aims to investigate the possible rhythmicity of the different components of the IGF-1 system (*igf-1*, the *igf1ra* and *igf1rb* receptors and the paralogs of its binding proteins IGFBP1 and IGFBP2) in the liver of goldfish. In addition, we also study the influence of two environmental cues, the light/dark cycle and feeding time, as *zeitgebers*. The hepatic *igf-1* expression showed a significant daily rhythm with the acrophase prior to feeding time, which seems to be strongly dependent on both *zeitgebers*. Only *igfbp1a-b* and *igfbp1b-b* paralogs exhibited a robust daily rhythm of expression in the liver that persists in fish held under constant darkness or randomly fed. The hepatic expression of the two receptor subtypes did not show daily rhythms in any of the experimental conditions. Altogether these results point to the *igf-1*, *igfbp1a-b*, and *igfbp1b-b* as clock-controlled genes, supporting their role as putative rhythmic outputs of the hepatic oscillator, and highlight the relevance of mealtime as an external cue for the 24-h rhythmic expression of the IGF-1 system in fish.

## 1. Introduction

The main endocrine pathway for animal growth is the somatotropic axis or the growth hormone-insulin-like growth factor-1 axis that comprises the hypothalamic control of pituitary somatotrophs and the consequent secretion of the growth hormone (GH). The main target for the GH is probably the liver, where it induces the synthesis and release of insulin-like growth factor-1 (IGF-1) that exerts negative feedback on pituitary synthesis and secretion of GH [1,2]. Several studies have focused on the GH, but only a few studies have investigated the three major components that constitute the IGF-1 signaling system, i.e., IGF-1, its receptors, and the binding proteins.

The IGF-1 is a single-chain polypeptide, originally called somatomedin C, that shares high structural homology with proinsulin and the IGF-2 [3], and it is well conserved in vertebrates. This growth factor is expressed in a wide variety of tissues, but the liver is the main producer of this peptide in most of the species studied so far [4]. The IGF-1 production is mainly controlled by the GH, but other hormones such as insulin and cortisol also participate [1,5], in addition to nutritional status, which is a key modulator in the control of IGF-1 [6]. The relevance of the IGF signaling system in growth regulation is currently accepted through vertebrates [7,8,9], and particularly in fish, IGF-1 has a remarkable pleiotropic profile as both an endocrine hormone and paracrine/autocrine growth factor [5,10]. It is reported to be involved in the stimulation of growth, cell survival and proliferation in different tissues, such as the retina, cartilage, muscle, and bone [11,12], regulation of food intake [13], osmotic adaptation to salinity changes [14,15], and reproduction [16,17], among others.

The bioactivity of IGF-1 requires its binding to specific receptors on the cell surface of targeted tissues activating well-established intracellular signaling transduction pathways [4,18]. The IGF-1 receptors (IGF1Rs) are characterized by a heterotetrameric arrangement consisting of two extracellular α-subunits that bind the ligand and two transmembrane β-subunits containing the tyrosine kinase domains [18]. The IGF-1 binding to the α-subunit activates the intrinsic tyrosine kinase activity, leading to its autophosphorylation on tyrosine residues as well as tyrosine phosphorylation of substrates involved in the intracellular signaling transduction pathways [18,19]. Due to the whole genome duplication 3R experienced by teleost [20], there are two isoforms of the IGF1Rs in most of them, the IGF1Ra and IGF1Rb. In addition, the extra round of genome duplication that occurred in both salmonid 4Rs [21] and the *Cyprininae* subfamily 4Rc, including goldfish (*Carassius auratus*) [22], is expected to produce four paralogs of these receptors. The broad tissue distribution of these receptors reported in some species [18,23,24,25] supports the multiple endocrine and paracrine actions reported for the IGF-1 in teleosts.

In vertebrates, the IGF-1 is not stored in tissues but is immediately secreted and bound to extracellular high-affinity insulin-like growth factor binding proteins (IGFBPs) [18]. The IGFBPs are single-chain polypeptides belonging to a family of cysteine-rich proteins highly conserved in vertebrates, including fish, that are essential for the regulation of systemic and local IGF-1 signaling [26]. In fact, the biological activity and half-life of IGF-1 are regulated by its high-affinity linkage to these IGFBPs, as in addition to carrying the circulating IGF-1, they protect the peptide from the enzymatic degradation and clearance [27], extending the half-life of IGF-1, and then, the availability of this peptide to its receptor, and consequently their actions. However, as most circulating IGF-1 is bound to IGFBPs due to the higher affinity of these binding proteins compared to the IGF-1 receptors, consequently, IGFBPs limit the availability of peptide for receptor activation. In any case, IGFBPs enable IGF-1 to act as an endocrine hormone and as autocrine/paracrine growth factors [28]. Six IGFBPs (−1 to −6 subtypes) have been described in mammals. The evolutive origin of this family is from a single pair of *igfbps* arranged in a tail-to-tail tandem organization in protochordates. Due to two rounds of whole genome duplications (1R and 2R) in ancestral gnathostomes, a total of eight *igfbp* paralogues are expected. However, two *igfbp* genes have been lost in all recent Chondrichthyes and Teleostomi [29]. The *igfbp* genes are widely expressed throughout the organism, with different expression patterns and regulation, and their physiological IGF-dependent and IGF-independent actions have been addressed in mammals [30] and fish [26]. In goldfish, due to the lineage-specific genome duplication events previously mentioned for the IGF-1 receptors (3R and 4Rc), the existence of up to four paralogs for each of the six IGFBP subtypes is expected. Members of the IGFBP family share sequence homology, but its regulation and the possibly distinct functional roles of this IGFBPs variety remain unsolved. Nevertheless, IGFBP1 and IGFBP2 have been reported as the main subtypes expressed in the liver of goldfish [31,32].

It is recognized that the actions of IGF-1 appear to be critically dependent on the hormonal and nutritional environment, but the relevance of the temporal organization of the IGF-1 axis has not been explored thoroughly. It is well known that a broad range of biological processes exhibit daily rhythmicity that has evolved to cope with natural environmental cyclic variations. This internal timing system confers adaptive advantages as it allows animals to anticipate and synchronize with periodic environmental variations [33]. In fish, studies on rhythms in the somatotropic axis have mainly focused on plasma GH. Daily rhythms of plasma IGF-1 in the channel catfish and liver *igf-1* mRNA expression in gilthead sea bream, rabbitfish, tilapia, zebrafish, and Senegalese sole have been described [34,35,36,37,38,39]. However, it is unknown whether these daily variations are exogenous or are entrained by predictable cues. In addition, very little is known about possible rhythmic variations in the rest of the key elements that constitute the IGF-1 system, i.e., the receptors and the IGFBPs.

The liver, besides its well-known metabolic role, is the main peripheral oscillator that exhibits strong molecular machinery of clock genes. This oscillator is mainly entrained by food availability, being considered a food-entrainable oscillator involved in the temporal orchestration of metabolic processes [40,41,42]. However, it remains to be explored if the hepatic IGF-1 system can be considered as a real output of the liver clock, and very little is known about the possible rhythmicity of the different components of this system and its feasible role in integrating environmental timing signals to orchestrate the signaling outputs of the liver clock. Goldfish represent a teleost model with a well-studied circadian system, and the chronobiological properties of the liver as a circadian clock are well-known [33,40,41,42]. However, there is a great lack of knowledge of its rhythmic outputs. The purpose of this study is to investigate the possible role of the IGF-1 system (including the IGF-1 peptide, its receptors, and the binding proteins 1 and 2) as an output of the liver clock using goldfish as an animal model.

## 2. Materials and Methods

### 2.1. Animals

Juvenile goldfish (*Carassius auratus*) with a body weight (bw) of 13.9 ± 0.4 g were obtained from a commercial supplier (Industrias Canarias del Acuario S.A., Madrid, Spain). Fish (*n* = 204) were maintained in 60-L tanks (*n* = 6–7 fish/tank) with filtered and aerated fresh water (21 ± 2 °C) and under controlled photoperiod 12 h light:12 h dark (12L:12D, lights on at 8.00 a.m., i.e., Zeitgeber Time 0, ZT0). They were fed once daily at 10.00 a.m. (ZT2) with a 1.5% bw ration of dry pellets (Sera Pond Biogranulat, Heinsberg, Germany). Goldfish were maintained under these conditions for at least 3 weeks before the experimental use, with a 100% survival during both acclimation and experimental periods. All experimental procedures complied with the Guidelines of the European Union Council (2010/63/UE) and the Spanish Government (RD53/2013) for the use of animals in research and were approved by the Animal Experimentation Committee of Complutense University of Madrid and the Community of Madrid (PROEX 170.6/20). The authors complied with the ARRIVE guidelines.

### 2.2. Tissue Distribution of the IGF-1 System in Goldfish

The interrenal gland, gills, heart, esophagus, intestinal bulb, anterior intestine, middle intestine, posterior intestine, spleen, liver, adipose tissue, kidney, gonads, skin and white muscle were collected from 6 fish at ZT4. The mRNA extraction, the cDNA synthesis, and the quantification of mRNA expression of *igf-1*, *igf1ra* and *igf1rb* by real-time quantitative PCR (RT-qPCR) were performed following the protocol below described (2.5. Gene expression analysis). The relative abundance of the 4 paralogs (a-a, a-b, b-b, and b-a) of *igfbp1* and *igfbp2* was quantified in the liver.

### 2.3. Daily Variations of the IGF-1 System in the Liver. Light/Dark Cycle and Scheduled Feeding as Zeitgebers

Fish were divided into 3 experimental groups and exposed to the following conditions for 30 days. (1) The standard conditions (SC) group consists of 42 fish maintained under 12L:12D photoperiod and daily fed at 10.00 a.m. (ZT2); (2) the 24 dark (24D) group, conducted for suppressing the light-dark cycle *zeitgeber*, and comprises 42 fish maintained under 24D darkness and daily fed at 10.00 a.m. (Circadian Time 2, CT2); (3) the random feeding (RF) group, carried out to suppress the feeding time *zeitgeber*, and consists of 42 fish maintained under 12L:12D photoperiod and randomly fed provided by the RAND function (Microsoft Excel^®^ software 16.0). In all experimental groups, food was provided by automatic feeders without altering either the light or dark conditions established at feeding time. The validity of this feeding protocol, which guarantees food intake within 30 min immediately after food delivery under 24D conditions, has been previously reported [43]. On the day of the experiment, fish were sampled every 4 h (*n* = 7 fish/sampling time) throughout a 24-h cycle (ZT3, ZT7, ZT11, ZT15, ZT19 and ZT23). For sampling, fish were anesthetized with tricaine methane sulfonate (MS-222, Sigma-Aldrich, St. Louis, MI, USA) and killed by anesthetic overdose (0.28 g/L). Liver samples were collected, immediately frozen in liquid nitrogen, and stored at −80 °C until RNA extraction.

### 2.4. Entrainment by the Feeding Time of the IGF-1 System in the Liver

Two experimental groups (*n* = 36 fish/group) were established. Fish were maintained for 30 days under 12L:12D photoperiod and fed at 2 different times, 1 group was scheduled to be fed in the middle of the light phase (ML, mid-light group, ZT6), and the other in the middle of the dark phase (MD, mid-dark group, ZT18). In both experimental groups, food was provided by automatic feeders without modifying the light or dark conditions established at feeding time. Fish fed at ZT 18 exhibited a robust food anticipatory activity during the night with a significant daily rhythm [44]. On the day of the experiment, fish from the two experimental groups were sampled every 4 h throughout the 24-h cycle (ZT5, ZT9, ZT13, ZT17, ZT21 and ZT25, *n* = 6 fish/sampling time). Fish were sacrificed by anesthetic overdose, and liver samples were collected, immediately frozen in liquid nitrogen, and stored at −80 °C until analysis.

### 2.5. Gene Expression Analysis

The number of transcripts of target genes (*igf-1*, *igf1rs*, and *igfbps*) was quantified by RT-qPCR, as described before [39]. Briefly, total RNA was extracted from tissues by using the guanidinium thiocyanate-phenol-chloroform reagent (TRI^®^Reagent, Sigma-Aldrich), according to manufacturer instructions with minor modifications. Aliquots of 0.3–3 µg of total RNA were treated with RQ1 RNase-Free DNase (Promega, Madison WI, USA) and subsequently retrotranscribed into cDNA (SuperScript Reverse II Transcriptase, Invitrogen, Walthman, MA, USA). The *β-actin*, elongation factor-1a (*ef-1a*) and 18S ribosomal RNA (*18S rRNA*) were used as reference genes to normalize quantified gene expression levels. The specific primer sequences used for targets and reference genes and their reference numbers (Gen Data Bank) are shown in Appendix A.

The RT-qPCR reactions were performed in a final volume of 10 µL, including 1 µL of cDNA and 0.5 µM of each forward and reverse primer, and using iTaqTM SYBR^®^ Green Supermix in a CFX96TM Real-Time System (BioRad Laboratories, Hercules, CA, USA). The qPCR cycling conditions consisted of a ramp of 95 °C for 30 s and 40 cycles of a 2-step amplification program (95 °C for 5 s and 60 °C for 30 s) for *igf-1*, *igfbps* and reference genes; and a ramp of 95 °C for 3 min and 40 cycles of a 3-step amplification program (95 °C for 10 s, 58 °C for 30 s and 60 °C for 30 s) for *igf1ra* and *igf1rb* genes. Calibration curves were made with serial dilutions of cDNA, showing efficiencies around 100%. The specificity of amplification was corroborated by melting curves (temperature gradient at 0.5 °C/5 s from 70 to 90 °C) at the end of each run and by the size of PCR products in a 1.5% agarose gel. Negative controls included the replacement of cDNA with water and the use of non-retrotranscribed RNA. The relative mRNA expression was determined by the 2^−ΔΔCt^ method [45], considering the lowest expression levels as the relative value of 1.

### 2.6. Statistics

Statistics were carried out using SigmaPlot 12.0 (Systat Software Inc., San José, CA, USA). Data were checked for normality (Shapiro-Wilk test) and homoscedasticity (Levene’s test). When necessary, data were log-transformed to fulfill the conditions of normality and homoscedasticity. Statistical differences in mRNA expression among sampling points were assessed using 1-way ANOVA, followed by the post hoc Student–Newman–Keuls (SNK) multiple comparison test. Differences were considered statistically significant at *p* < 0.05. The daily rhythms of gene expression were assessed by Cosinor analysis by fitting gene expression values throughout the 24 h to periodic sinusoidal functions s by the least squares method [46]. The formula used was f(t) = M + A·cos(t·π/12−ϕ), where f(t) is the gene expression level at a given time point, the mesor (M) is the middle value of the fitted cosine representing a rhythm-adjusted mean, A is the sinusoidal amplitude of oscillation (i.e., the half the difference between the minimum and maximum of the fitted cosine function), t is time in hours and ϕ is the acrophase (time of the peak value expressed in h from ZT0). The estimation of M, A, and ϕ and their standard error (SE) was carried out using nonlinear regression. The significance of the cosinor analysis was tested using the zero-amplitude test, which indicates if the sinusoidal amplitude differs from 0 with a given probability [47]. Daily rhythms were significant when both *p* < 0.05 by ANOVA and *p* < 0.01 by the zero-amplitude test were achieved.

## 3. Results

### 3.1. Tissue Distribution of the IGF-1 System in Goldfish

The relative gene expression of the main components of the IGF-1 system in the peripheral tissues of goldfish is shown in Figure 1. The highest abundance of *igf-1* mRNAs was found in the liver and adipose tissue, and low expression was found in the rest of the tissues, except for the interrenal tissue, heart, kidney, gonads, skin, and muscle, where very low expression of *igf-1* was found (Figure 1A). Regarding the distribution of IGF-1 receptors, the *igf1ra* subtype shows a widespread tissue distribution with similar abundance in most of the studied tissues and the highest expression values in the gills and heart and the lowest expression found in the middle intestine, liver and muscle (Figure 1B). The transcripts of *igf1rb* were abundantly located in gills and gonads, with the liver and muscle as the tissues that exhibited the lowest expression (Figure 1C). The analysis of the relative abundance of the *igfbp1* and *igfbp2* paralogs in the liver of goldfish showed the highest expression values for the *igfbp1b-a* and lower expression of the *igfbp1a-b*. The *igfbp1a-a* and *igfbp1b-b* and the four IGFBP2 paralogs were scarcely expressed in the liver of goldish (Figure 1D).

### 3.2. Daily Variations of the IGF-1 System in the Liver. Light/Dark Cycle and Scheduled Feeding as Zeitgebers

The *igf-1* expression in the liver of goldfish showed a significant 24-h daily rhythm (ANOVA and cosinor analysis) of low amplitude (A = 0.55) in animals under a 12L:12D photocycle and fed at ZT2 (2 h after lights on). The acrophase of this rhythm is found at the end of the dark period (ZT23; Figure 2A). This daily rhythmic expression disappeared in the absence of external *zeitgebers*, the light/dark cycle (24D group, Figure 2B) or the scheduled feeding (RF group, Figure 2C). The mean levels of expression throughout the 24-h remained similar in both groups compared to fish exposed to both *zeitgebers* (SC group).

Liver expression of the two paralogs of *igfbp1* (*igfbp1a-b* and *igfbp1b-b*) exhibited robust daily rhythms in goldfish under a 12L:12D photocycle and daily fed at ZT2, with the acrophase found at mid-photophase (ZT6.7, 4-h post-feeding), and the nadir at mid-dark for the *igfbp1a-b* (Figure 3A). The amplitude of the daily rhythm for the *igfbp1b-b* paralog was lower than the amplitude of *igfbp1a-b*, and the rhythm was slightly shifted, with the acrophase placed around ZT10 and the nadir at the end of scotophase (Figure 3B). These sinusoidal rhythms were preserved in goldfish reared under 24D, but the daily profile of expression of both binding proteins paralogs was shifted (6-h for *igfbp1a-b* and 3 h for *igfbp1b-b*), while the amplitude remained almost unaltered for *igfbp1a-b* (Figure 3C) but resulted slightly reduced for the *igfbp1b-b* expression (1.45 versus 2.65, Figure 3D). In the absence of the scheduled feeding as a *zeitgeber*, the daily rhythm of the *igfbp1a-b* transcripts preserved their 24-h profile and amplitude, and the acrophase slightly shifted (ZT8, Figure 3E). The expression pattern of the *igfbp1b-b* paralog remained similar to that found in 24D, with a shift of 6 h compared to the control group (Figure 3F). The gene expression of the rest of the paralogs of IGFBP1 and the four paralogs of IGFBP2 did not show statistically significant variations throughout the 24-h.

The 24-h expression of both subtypes of IGF-1 receptors (*igf1ra* and *igf1rb*) in the liver remained unmodified in fish exposed to both *zeitgebers* (light/dark cycle and scheduled feeding time group, Figure 4A,B). Similar results were found in the absence of the light/dark cycle (24D group, Figure 4C,D) or a scheduled feeding (randomly fed group, Figure 4E,F), except for the expression of *igf1rb* in the 24D group, where the abundance of transcripts was significantly higher at CT15 than at CT7 (Figure 4D), but data were not significant by cosinor analysis

### 3.3. Entrainment by the Feeding Time of the IGF-1 System in the Liver

The possible entrainment of the hepatic rhythm of *igf-1* by feeding time was studied by a 12h-shift of feeding time in goldfish under 12L:12D photoperiod. A daily rhythm of *igf-1* expression in the liver was found in both groups of goldfish, with similar amplitudes and mesor values, but the acrophases were 12-h shifted in fish fed at mid-dark (MD group, ZT18, Figure 5B) compared to fish fed at midday (ML group, ZT6, Figure 5A).

The effects of a 12 h-shift in feeding time on the expression of *igfbp1* and *igfbp2* paralogs are shown in Figure 6 and Figure 7, respectively. The expression of the *igfbp1a-b* and *igfbp1b-b* paralogs showed significant daily rhythms of similar amplitudes that shifted when the zeitgeber feeding time was 12h-shifted (Figure 6C,D,G,H). It is found that the amplitude of these rhythms was higher in fish entrained by feeding time at mid-dark compared to fish fed at midday. The daily expression of *igfbp1a-a* in fish fed at midday showed two peaks: at the end of photophase and at the end of scotophase, this profile is suitable with an ultradian rhythm (Figure 6A), but it is modified to a low amplitude 24-h sinusoidal rhythm when food was provided at the mid-dark (Figure 6B). The expression of *igfbp1b-a* showed statistically significant differences throughout the 24-h (ANOVA), but it did not fit sinusoidal rhythmic waves in any of the two mealtime conditions, midday (Figure 6E) and mid-dark (Figure 6F).

As a rule, the expression of the four paralogs of the *igfbp2* did not fit sinusoidal 24-h rhythms, but some statistically significant differences were found through the 24-h cycle (Figure 7). The number of transcripts of *igfbp2a-b* and *igfbp2b-b* was significantly higher at the beginning of the light phase in fish fed at midday (Figure 7A,G), but this pattern disappeared in fish fed at mid-dark (Figure 7B,H).

Figure 8 shows polar graphs with the amplitudes and acrophases derived from the cosinor analysis for the daily rhythms in *igf-1* (A), *igfbp1a-b* (B) and *igfbp1b-b* (C) expression in the liver. The outermost ring indicates the time in *zeitgeber* units, and the acrophase (ϕ) is indicated by the angle of a vector whose length corresponds to the amplitude (A). It is clearly observed the 12-h shift in the daily rhythmic profile of *igf-1* expression without alterations in amplitudes (Figure 8A). In the case of the *igfbp1a-b* (Figure 8B), it can be observed the amplitude changes induced by different feeding schedules and the shifted acrophases for the rhythmic profiles placed in the light phase of the photocycle (except for the rhythm under continuous darkness). Finally, in the random feeding group, a reduction of amplitudes and the shift of acrophases from dusk to the first half of the dark phase in the *igfbp1b-b* rhythms are shown in Figure 8C.

## 4. Discussion

Present data provide relevant information about the IGF-1 system as a new candidate among the possible functional outputs of the liver clock in fish. Our study reveals that the IGF-1 in peripheral tissues of goldfish is mostly produced in metabolic tissues, such as the liver and adipose tissue, with significantly lower presence in other tissues, such as the gills and gastrointestinal tract. Very few studies report the presence of IGF-1 in adipose tissue, but the abundance of this peptide in the liver is a common feature in all the teleost species so far investigated (coho salmon [48,49], tilapia [50], zebrafish [51], common carp [52], trout [53], and silver pomfret [54], including the goldfish [12]) and point to the liver as the primary source of IGF-1 in fish. It is important to note that this pattern of tissue distribution could be slightly different depending on the time of day it is quantified due to the daily variation of *igf-1* expression found in the liver. The very low expression of both subtypes of IGF-1 receptors found in the liver of goldfish, in agreement with previous reports in other teleosts [25,51,55], supports that IGF-1 from the liver is released into the blood without competition from IGF1Rs, evidencing the endocrine role of hepatic IGF-1 [56]. On the other hand, our results about a broad tissular distribution of *igf1ra* and *igf1rb*, together with the presence of the *igf-1* transcripts in such a wide variety of tissues, support the pleiotropic profile of this peptide in fish and reinforce the hypothesis that the local production of this peptide may exert paracrine/autocrine actions in multiple organs in goldfish, as suggested in other vertebrates [18]. Liver and muscle exhibited negligible gene expression of both IGF1R subtypes. Meanwhile, gills were the tissue with the highest expression of both *igfr1a* and *igfr1b* receptors, in agreement with the suggested osmoregulatory functions of IGF-1 in fish [15,57]. The abundant expression of the *igf1rb* subtype in the gonads also supports the reproductive functions of IGF-1 in fish [17,54]. Few studies have reported differential tissular distribution of different subtypes of IGF-1 receptors in fish [25,51,55], with similar results to those shown in the present study.

Our results show that both IGFBP1 and IGFBP2 encoding genes are expressed in the liver of goldfish and support the synthesis of these proteins in this tissue [31,32]. It is generally accepted that the liver is the predominant source of circulating IGFBPs in different teleosts [26], e.g., the liver exhibits a higher abundance of IGFBP transcripts than any other tissue in the Atlantic salmon, 95% of which comprised IGFBP1 and IGFBP2 paralogs [58]. In addition, our study demonstrates for the first time the gene expression of the four paralogs of each of the two IGFBPs (IGFBP-1 and IGFBP-2) in the liver of goldfish, which we have named *igfbp1a-a*, *igfbp1a-b*, *igfbp1b-a*, *igfbp1b-b*, and *igfbp2a-a*, *igfbp2a-b*, *igfbp2b-a*, and *igfbp2b-b*, based on the information from the syntenic analysis (in preparation). This nomenclature agrees with the suggestion for *Cyprinus carpio* (α = a, β = b) but differs from that used for salmonids, whose *igfbp* genes are not orthologous due to an independent genome duplication [26]. The most abundant paralog expressed in the liver of goldfish is *igfbp1b-a*, while the lower expression is found for the *igfbp2b-a* paralog. Many studies do not distinguish paralogs of these *igfbp*-encoding genes, and the only one published in goldfish classified it into two groups, IGFBP-1a and IGFBP-1b, and grouped IGFBP-1 into the IGFBP-1a clade [31], which is probably an artefactual chimera of IGFBP1a-a and IGFBP1a-b. Also, in zebrafish, it has been identified two genes that are co-orthologs of human IGFBP-1, *igfbp-1a* and *igfbp-1b*, whose expressions are restricted to the liver of adults [59]. Despite a wide tissular distribution of IGFBP-2 reported in teleosts, with higher abundance in the liver and the nervous system [26], the analysis of paralogs shows the undetectable expression of *igfbp2b2* in the liver of Atlantic salmon [58], consistent with our data in goldfish, and emphasize the relevance of distinguishing paralogs to provide insights into the functional roles of the different members of the IGFBPs family.

The liver of goldfish exhibits daily rhythms in the transcripts of some elements of the IGF-1 system, particularly IGF-1 and some IGFBPs, supporting the hypothesis that the IGF-1 system could be acting as a rhythmic output of this peripheral clock. The daily rhythm in the abundance of *igf-1* transcripts found in the liver of goldfish with the acrophase at the end of the dark period matched with daily rhythms of clock genes belonging to the negative limb (e.g., *per* genes) of the main loop of molecular machinery in the liver clock [60]. This result points to *igf-1* as a clock-controlled gen and reinforces its possible role as a rhythmic output of the liver oscillator. In support of this proposal, it is reported that the promoter of the IGF-1 encoding gene in the white seabream and the common carp contains highly conserved regulatory elements [61,62], some of which are identified as transcription factors in the circadian regulation of the clock-controlled genes in mammals [63].

Daily variations of *igf-1* mRNA expression have been reported in some teleost [35,37,38], but significant circadian rhythms were demonstrated only in the Senegalese sole [36], the gilthead sea bream larvae [39] and the Nile tilapia [62]. By contrast, no daily pattern was observed in the expression of *igf-1* mRNA in rabbitfish through the light/dark cycle [34]. This variability in the daily profile of *igf-1* expression may be due to species differences (e.g., diurnal versus nocturnal, larvae versus adults), but in chronobiological studies, it is essential to properly define the environmental conditions (photocycle, feeding protocol, activity pattern) during the study, because these factors clearly entrain daily rhythms [33,40,64]. In agreement with our data, in the Senegalese sole [36] and the tilapia [65] under a 12L:12D photocycle, the acrophases of *igf-1* daily rhythms were located toward the end of the dark phase of the photocycle, which has been suggested as the anticipation of light onset. Our study demonstrates that a photocycle is required to express the daily rhythm of *igf-1* in the liver of goldfish, as this rhythm disappears under 24D conditions, as occurs in tilapia [65], and supports previous reports that evidence the relevance of photoperiod for the expression of *igf-1* daily variations in the liver of teleost [35,36,37,39]. The fact that clock genes expression (*per1*, *clock1*, *bmal1*, and *rev-erb*) in the liver of goldfish was highly sensitive to the daily photocycle [40,60] demonstrates the key role of the light/dark cycle in the synchronization of this peripheral clock [33,41]. However, the lack of a photocycle does not abolish the daily rhythms of the two rhythmic paralogs of *igfbp1* in the liver of goldfish, which maintained significant rhythms with lower amplitudes and shifted the acrophases. This persistence of the *igfbp* rhythms in fish held under constant darkness could suggest that they were driven by an endogenous biological clock. Nevertheless, of the eight *igfbps* paralogs studied in goldfish, only two (*1a-b* and *1b-b*) exhibit a daily rhythm of expression in the liver of goldfish, which supports the distinct functional roles of this *igfbps* repertoire. The regulatory systems controlling the production of IGFBPs are largely unknown, and particularly, daily changes in the expression of IGFBP encoding genes and their regulation by the photoperiod have not been previously studied in the liver of fish. Results from this study in goldfish suggest that at least the *igfbp1a-b* and *igfbp1b-b* paralogs in the liver may be considered putative clock-controlled genes. The finding that the HIF-1, a candidate in the circadian transcriptional regulation of the clock-controlled genes in mammals [63], mediates hypoxia-induced IGFBP-1 gene expression in zebrafish early development support this idea. Also, in the skeletal muscle of zebrafish adults, it is suggested that *igfbp3* and *igfbp5* might be considered putative clock-controlled genes [66].

Even though the expression of IGF-1 receptors was very low in the liver of goldfish, we investigated possible changes in the expression through the 24-h cycle. We have not detected significant variations in the expression of any of the two receptor subtypes, the *igf1ra* and *igf1rb*, in any of the experimental conditions, except for a significant difference in the amount of *igf1rb* transcripts in fish under 24D, without any rational explanation for this finding at present. To our knowledge, few studies reported daily rhythms of IGF-1 receptors in fish, with a variety of results. In the Senegalese sole, a weak rhythm is described for *igf1r* with the acrophase close to midday [36], but in the Nile tilapia, no expression of *igf1ra* could be detected in the liver [35], but a clear daily rhythm in the liver transcripts of this receptor is recently reported in this teleost under 12L:12D, that disappears under 24D [65]. From these results, it seems that the light/dark cycle is important for the daily expression of *igf1rs* in the liver, but differences in feeding pattern (once a day at midday versus twice a day, at the beginning and the end of the light period) could explain the different results.

Light has been traditionally considered the main *zeitgeber* for the circadian system, but feeding time is a potent synchronizer of peripheral clocks, as the liver, which is considered a food-entrained oscillator [33]. Little is known about the functional organization of such clocks, but it is demonstrated that the liver drastically alters its molecular clockwork by different exogenous and endogenous inputs [33,41,60]. Our data clearly demonstrate the key role of feeding times in the entrainment of *igf-1* rhythms in the liver. On the one hand, the rhythm is abolished under random feeding conditions; on the other, a 12-h shift in feeding time induces a 12-h shift in the acrophase of the *igf-1* daily rhythm. Both responses are consistent with the changes of clock genes expression to this challenge of a 12-h shift of feeding time as synchronizer [44] and indicate that the IGF-1 encoding gene in the liver can be considered as a possible output gene of this peripheral clock, supported by a network between the functioning of the molecular machinery of the liver clock in goldfish and one of its putative outputs.

The acrophase of *igf-1* daily rhythms is always found around 4 h prior to feeding time in all the scheduled feeding experiments in the liver of goldfish, i.e., at ZT23, ZT2, and ZT13 when fish were fed once a day at ZT2, ZT6, and ZT18, respectively. The important finding that the 12-h shift in feeding time (from midday to midnight) shifted the acrophase of *igf-1* transcripts demonstrates that a scheduled feeding time entrains the *igf-1* daily rhythm in the liver of this teleost and supports the key role of the liver as a food-entrained clock in the functional circadian system of fish. In some teleosts, the peak of *igf-1* expression in the liver when fish are schedule-fed once a day during the light time is found at postprandial times: 5–6 h in rabbitfish [34] and 10 h in gilthead seabream [37]. These differences may be due to species differences, but they can also be related to the active phase of the animals. For example, in a nocturnal teleost, the Senegalese sole, the acrophase of *igf-1* was in the second half of the dark period of the photocycle [36]. In addition, in some teleosts, it is demonstrated that food availability, as a powerful synchronizer of endogenous clocks, drives locomotor activity [67,68]. Particularly in goldfish, the existence of a functional food-entrainable oscillator regulating locomotor activity is proposed on the basis that the scheduled feeding entrains an anticipatory food activity, even in total darkness, that is abolished under random feeding conditions [43].

The fact that the rhythmic profile of *igfbp1a-b* expression remains unaltered in random-fed fish might suggest that the daily rhythm of this paralog is independent of a scheduled feeding time as a *zeitgeber*. However, the results obtained in fish fed at different times (ZT2, ZT6 and ZT 18) seem to indicate that feeding time entrains the rhythms of this paralog, as the acrophase is found around 4-h post-feeding when food is provided during the light time, and 8-h when is provided during darkness. Taking in mind the relevance of the masking effects in chronobiological studies, from our results, we cannot conclude that the observed rhythms of the *igfbp1a-b* transcripts were really driven by an endogenous clock, as they can be the result of a passive consequence of a particular environmental condition, as the feeding time. Our results about the rhythmic expression of the *igf1bp1b-b* were slightly different from those obtained for the *igfbp1a-b* paralog, as the daily rhythms were preserved (with some delays in acrophases) in the absence of each one of the two zeitgebers (light/cycle and scheduled feeding time), but the profile exhibited significantly higher amplitude and phase delay when goldfish were fed at midnight. We have no explanation for this response, but considering that IGFBPs are multifunctional proteins with many different IGF-1-independent actions and are particularly involved in metabolic homeostasis [26], it is plausible to hypothesize that the expression of this paralog would be affected by a pattern of nocturnal feeding and the possible consequent alteration in homeostasis. From a chronobiological point of view, the liver expression of *igfbp1a-a* and *igfbp1b-a* of goldfish showed daily variations but not significant rhythms, and their specific role in the circadian system remains uncertain. Finally, the modulatory role of nutritional status on the somatotropic axis is well-known, but there is no information available on its possible effects on the daily rhythms of the IGF-1 system in fish. Nevertheless, the failure to detect a daily rhythm of *igf-1* in the rabbitfish [34] could be due to the fact that fish were 72-h fasted prior to sampling, as it is reported that plasma IGF-1 and liver *igf*-expression typically decline during food deprivation [2,69,70,71,72]. Thus, it could be suggested that nutritional status also plays an important role in endogenous signaling in the modulation of the rhythms of the IGF-1 system, which may act as an output of the liver clock.

Altogether these results encourage us to investigate whether IGF-1 is not only an output of the liver clock but could also act as a functional input in the crosstalk among clocks. Indeed, it could be essential to identify the expression of IGF-1 receptors in the brain and peripheral clocks, as well as to delve into the putative role of IGF-1 as a signal involved in the interplay among oscillators.

## 5. Conclusions

Present data provide relevant information about the IGF-1 system as a new candidate among the possible functional outputs of the liver clock in fish. Our results about the local production of the IGF-1 and the presence of receptors in different tissues support its paracrine/autocrine actions in goldfish. We demonstrate for the first time the gene expression of the four paralogs of each of the two IGFBPs (IGFBP-1 and IGFBP-2) in the liver of goldfish, which we have named *igfbp1a-a*, *igfbp1a-b*, *igfbp1b-a*, *igfbp1b-b*, and *igfbp2a-a*, *igfbp2a-b*, *igfbp2b-a* and *igfbp2b-b*. Such a wide variety of observed paralogs encourages the address of specific approaches to provide insights into the functional roles of the different members of the IGFBPs family.

Our data clearly demonstrate the key role of feeding times in the entrainment of daily rhythms in transcripts of hepatic IGF-1 and support the liver as a food-entrained clock. In addition, the *igf-1* and, at least, the *igfbp1a-b* and *igfbp1b-b* paralogs seem to be putative clock-controlled genes, reinforcing the hypothesis that the IGF-1 system may be acting as a rhythmic output of a food-entrainable oscillator, as it is the liver of goldfish.

## Figures and Tables

**Figure 1 animals-12-03371-f001:**
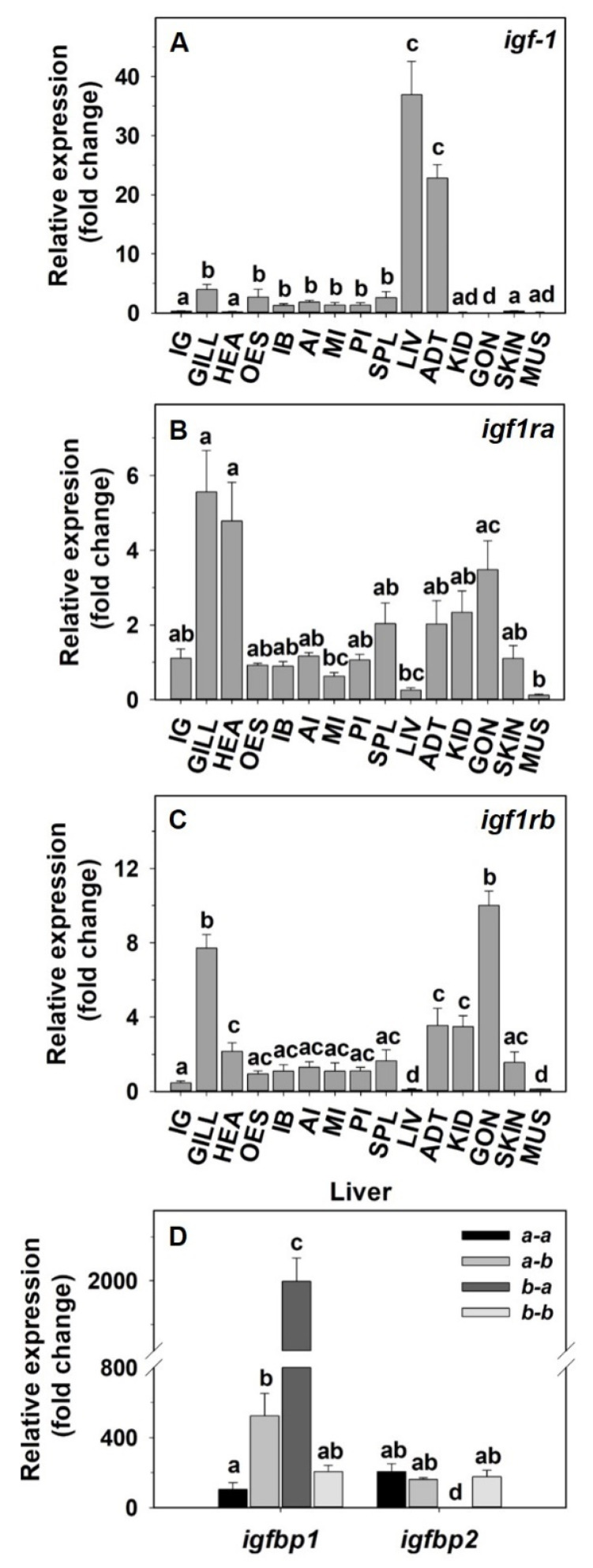
Tissue distribution of the IGF-1 system in peripheral tissues of goldfish. Relative expression of (**A**) *igf-1*, (**B**) *igf1ra*, (**C**) *igf1rb*, (**D**) the four paralogs (a-a, a-b, b-a, b-b) of *igfbp1* and *igfbp2* in the liver. Data are expressed as mean ± s.e.m. (*n* = 6) relative to the posterior intestine (*igf****-****1*, *igf1ra* and *igf1rb*). Different letters indicate statistical differences among tissues or subtypes. Interrenal gland (IG), gill (GILL), heart (HEA), esophagus (OES), intestinal bulb (IB), anterior intestine (AI), middle intestine (MI), posterior intestine (PI), spleen (SPL), liver (LIV), adipose tissue (ADT), kidney (KID), gonad (GON), skin (SKIN), muscle (MUS).

**Figure 2 animals-12-03371-f002:**
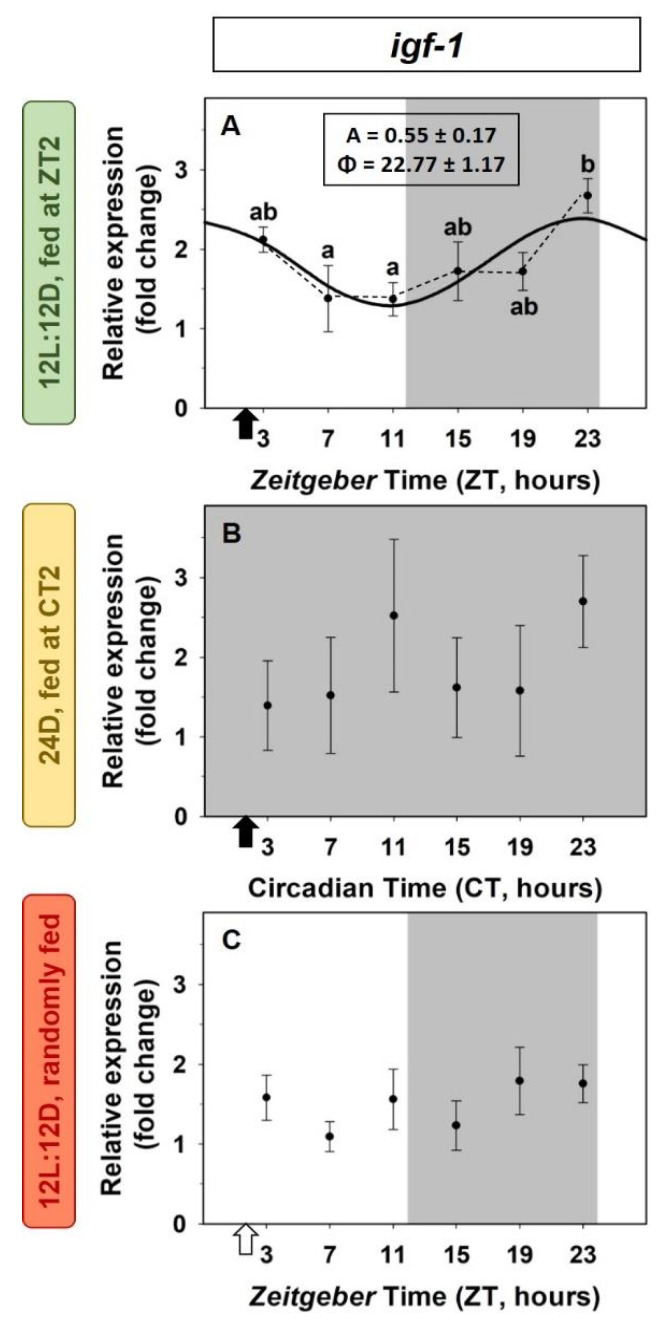
Daily variations of *igf-1* expression in the liver of goldfish. (**A**) standard conditions group (12L:12D photoperiod, feeding time at ZT2); (**B**) 24D group (24 h darkness, feeding time at CT2); (**C**) random feeding group (12L:12D photoperiod, randomly fed). Data are expressed as mean ± s.e.m. (*n* = 7/sampling time). Different letters indicate statistical differences among sampling times (ANOVA). Sinusoidal waves represent significant rhythm by cosinor analysis. Dashed lines represent significant differences among sampling points by ANOVA. Black arrows indicate feeding time. The white arrow indicates the time of the last feeding. The dark period is represented in grey. Parameters defining the rhythms (A: amplitude, ϕ: acrophase).

**Figure 3 animals-12-03371-f003:**
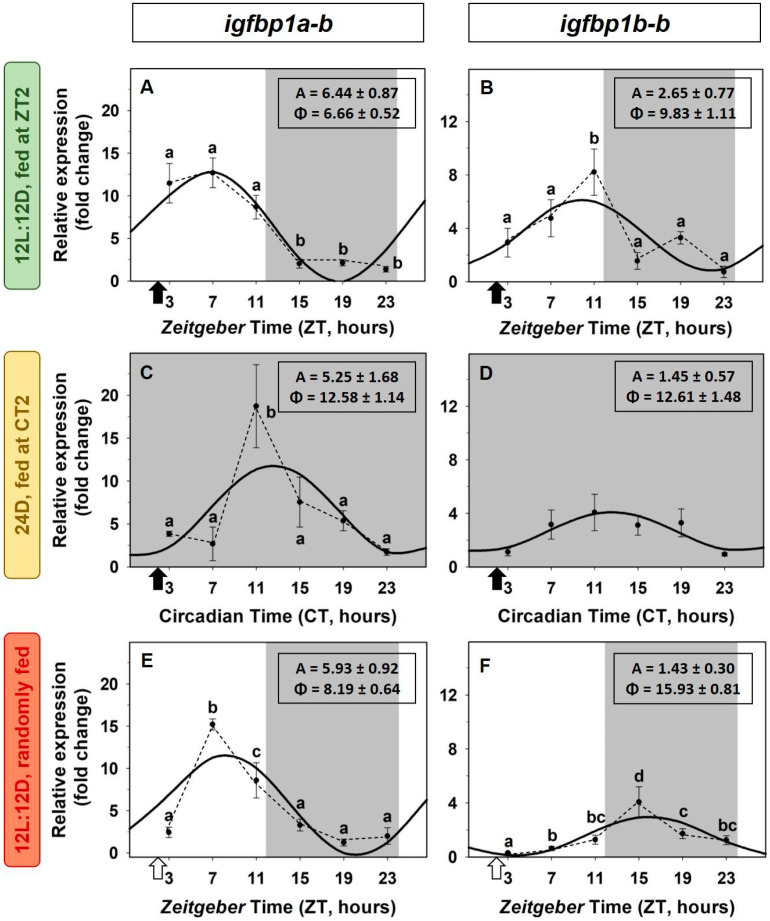
Daily variations of *igfbp1a-b* (**A**,**C**,**E**) and *igfbp1b-a* (**B**,**D**,**F**) expression in the liver of goldfish. (**A**,**B**) Standard conditions group (12L:12D photoperiod, feeding time at ZT2); (**C**,**D**) 24D group (24 h darkness, feeding time at CT2); (**E**,**F**) random feeding group (12L:12D photoperiod, randomly fed). Data are expressed as mean ± s.e.m. (*n* = 7/sampling time). Different letters indicate statistical differences among sampling times (ANOVA). Sinusoidal waves represent significant rhythms by cosinor analysis. Dashed lines represent significant differences among sampling points by ANOVA. Black arrows indicate feeding time. White arrows indicate the time of the last feeding. The dark period is represented in grey. Parameters defining the rhythms (A: amplitude, ϕ: acrophase).

**Figure 4 animals-12-03371-f004:**
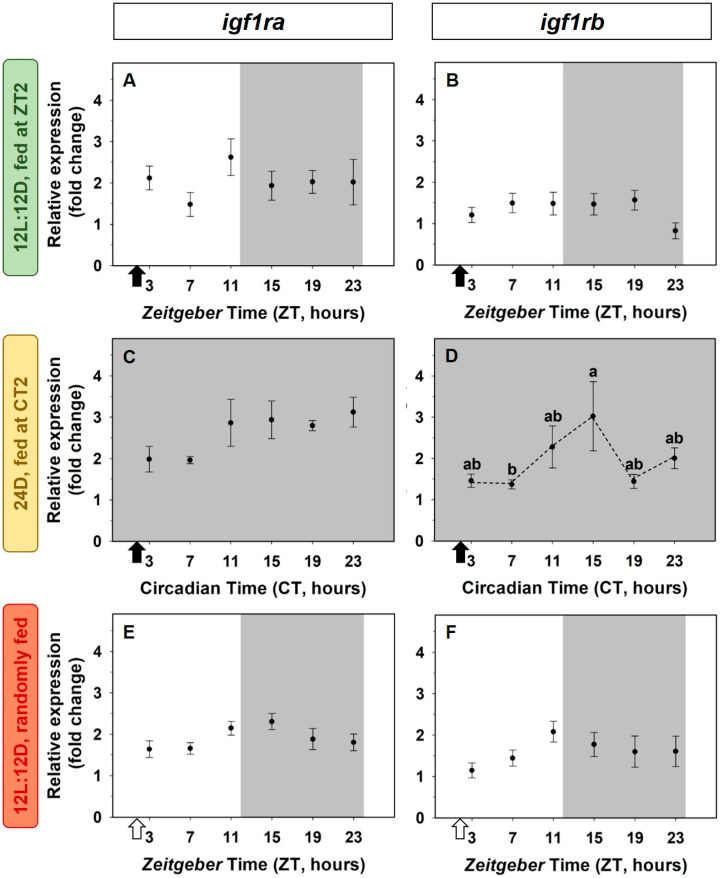
Daily variations of *igf1ra* (**A**,**C**,**E**) and *igf1rb* (**B**,**D**,**F**) expression in the liver of goldfish. (**A**,**B**) Standard conditions group (12L:12D photoperiod, feeding time at ZT2); (**C**,**D**) 24D group (24 h darkness, feeding time at CT2); (**E**,**F**) random feeding group (12L:12D photoperiod, randomly fed). Data are expressed as mean ± s.e.m. (*n* = 7/sampling time). Different letters indicate statistical differences among sampling times (ANOVA). Dashed lines represent significant differences among sampling points by ANOVA. Black arrows indicate feeding time. White arrows indicate the time of the last feeding. The dark period is represented in grey.

**Figure 5 animals-12-03371-f005:**
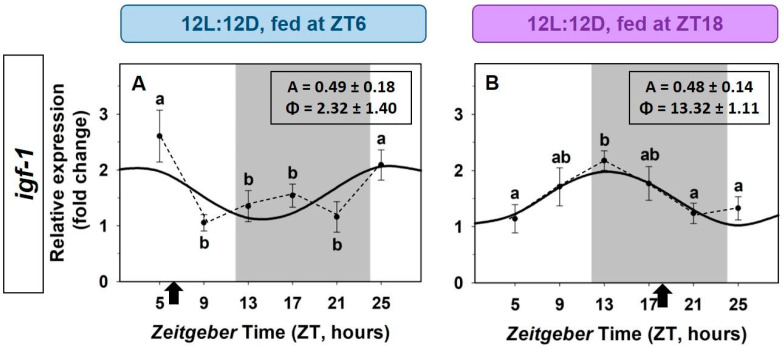
Effect of a shifted mealtime on the daily expression of *igf-1* in the liver of goldfish. (**A**) ML group (fish fed at the mid-photophase, ZT6). (**B**) MD group (fish fed at mid-scotophase, ZT18). Data are expressed as mean ± s.e.m. (*n* = 6/sampling time). Different letters indicate statistical differences among sampling times (ANOVA). Sinusoidal waves represent significant rhythms by Cosinor analysis. Dashed lines represent significant differences among sampling points by ANOVA. Black arrows indicate feeding time. The dark period is represented in grey. Parameters defining the rhythms (A: amplitude, ϕ: acrophase).

**Figure 6 animals-12-03371-f006:**
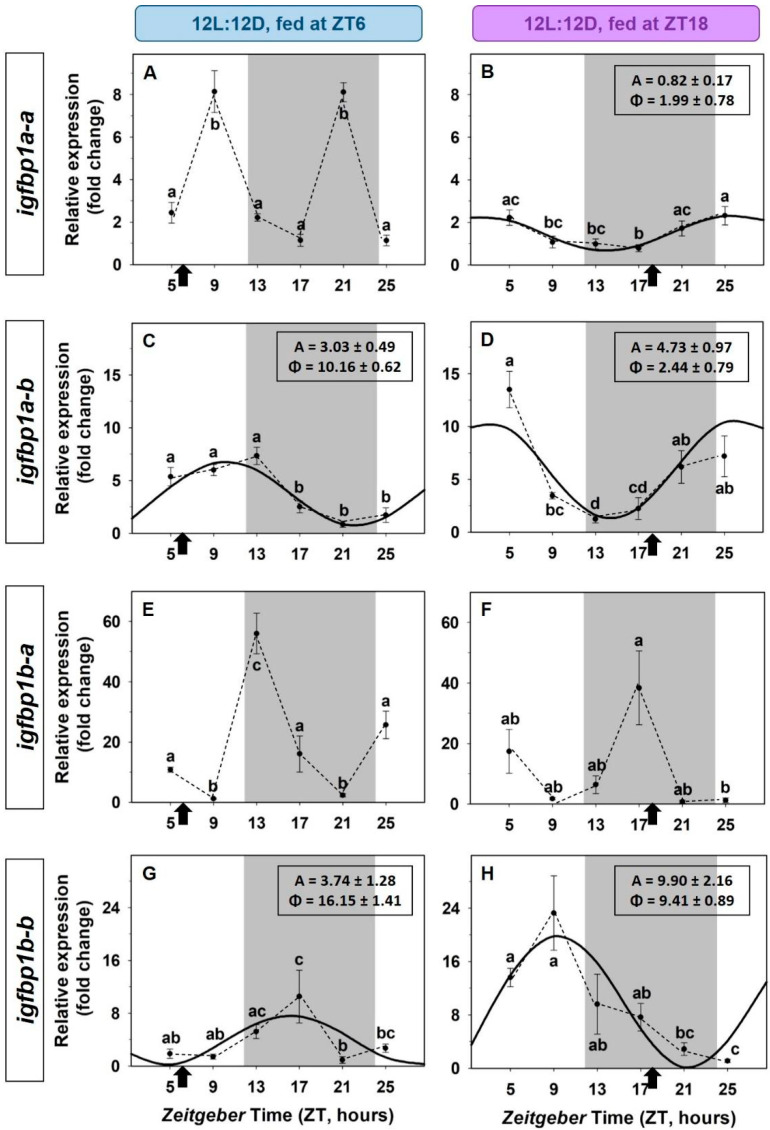
Effect of a shifted mealtime on the daily expression of *igfbp1* paralogs in the liver of goldfish. (**A**,**B**) *igfbp1a-a*; (**C**,**D**) *igfbp1a-b*; (**E**,**F**) *igfbp1b-b*; (**G**,**H**) *igfbp1b-a*. On the right (**A**,**C**,**E**), fish fed at mid-photophase, ZT6); on the left (**B**,**D**,**F**), fish fed at mid-scotophase, ZT18). Data are expressed as mean ± s.e.m. (*n* = 7/sampling time). Different letters indicate statistical differences among sampling points (ANOVA). Sinusoidal curves represent significant rhythms by Cosinor analysis. Dashed lines represent significant differences among sampling points by ANOVA. Black arrows indicate feeding time. The dark period is represented in grey.

**Figure 7 animals-12-03371-f007:**
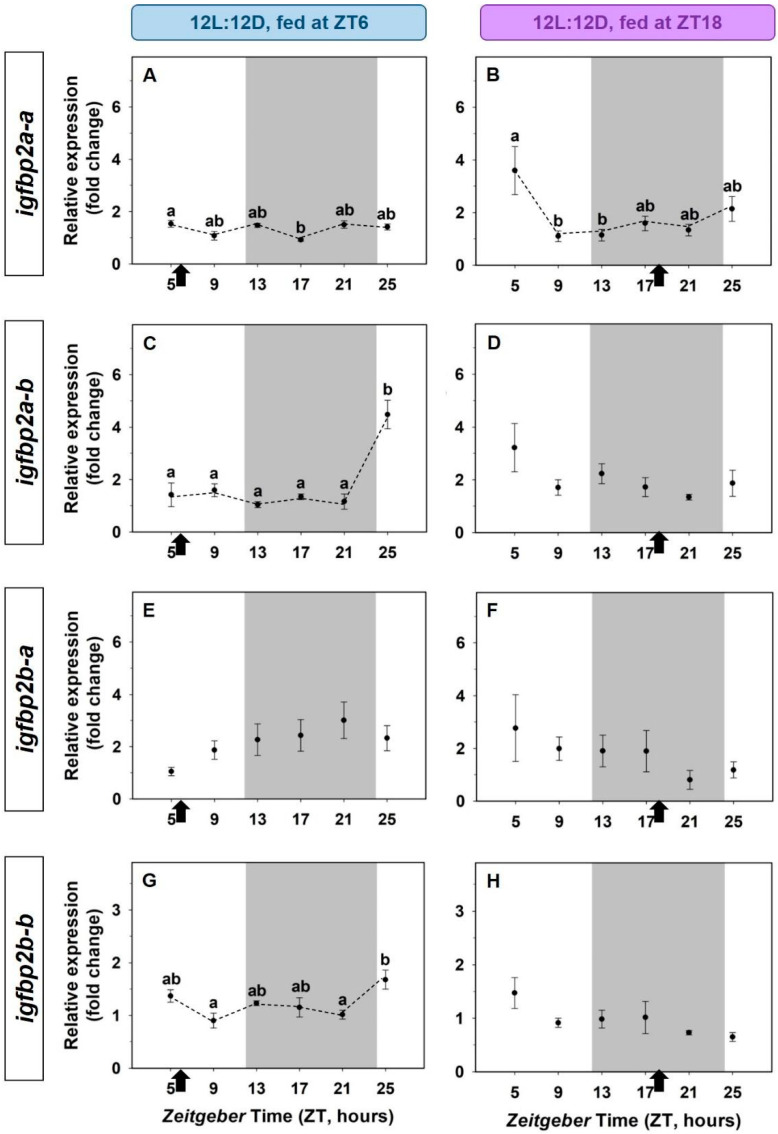
Effect of a shifted mealtime on the daily expression of *igfbp2* paralogs in the liver of goldfish. (**A**,**B**) *igfbp2a-a*; (**C**,**D**) *igfbp2a-b*; (**E**,**F**) *igfbp2b-b*; (**G**,**H**) *igfbp2b-a*. On the right (**A**,**C**,**E**), fish fed at mid-photophase, ZT6); on the left (**B**,**D**,**F**), fish fed at mid-scotophase, ZT18. Data are expressed as mean ± s.e.m. (*n* = 7/sampling time). Different letters indicate statistical differences among sampling points (ANOVA). Dashed lines represent significant differences among sampling points by ANOVA. Black arrows indicate feeding time. The dark period is represented in grey.

**Figure 8 animals-12-03371-f008:**
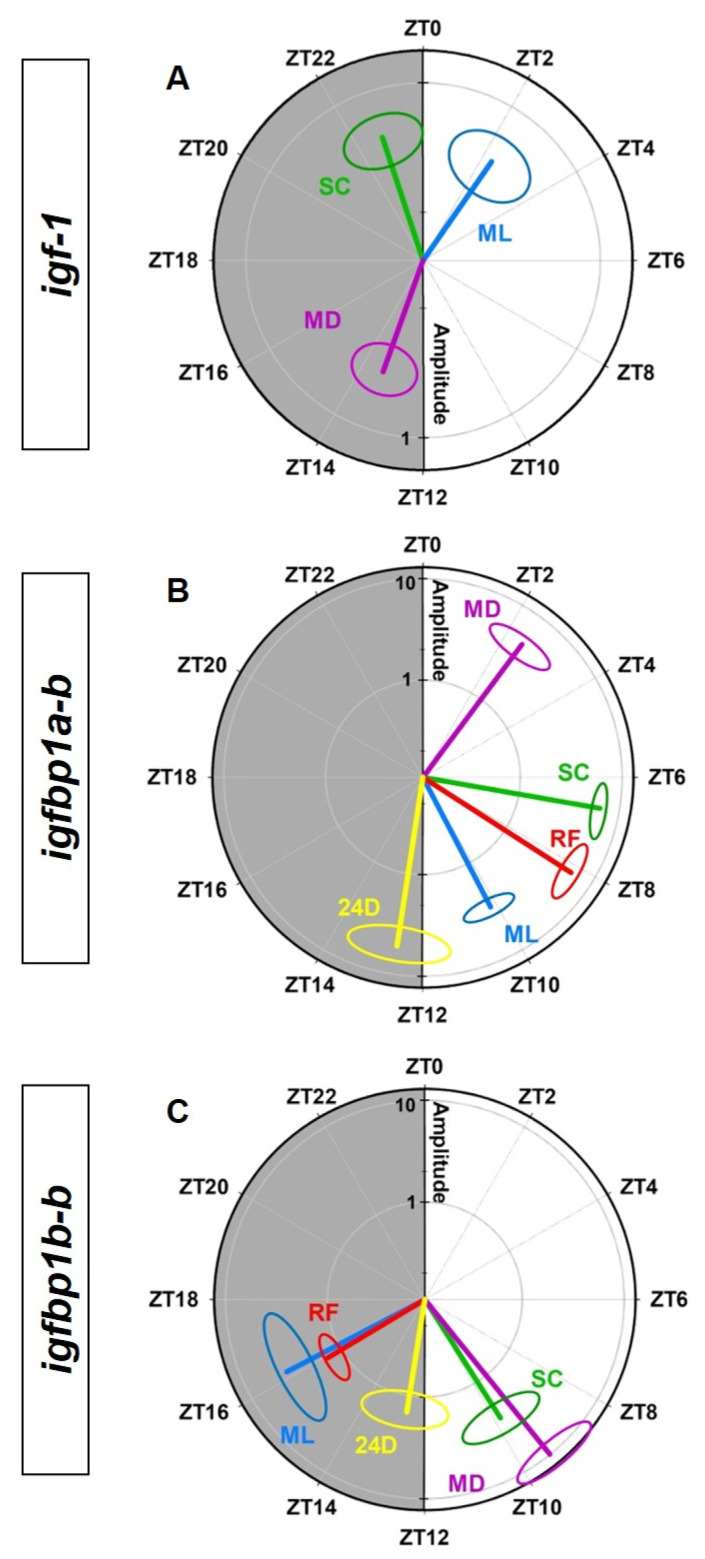
Polar representations of cosinor-derived rhythmic parameters of *igf-1* (**A**), *igfbp1a-b* (**B**) and *igfbp1b-b* (**C**) gene expressions under the different experimental conditions. Amplitude and acrophase are represented by a single vector starting from the center. The length of the vector (radial axis) indicates the magnitude of the amplitude (fold change of relative expression,) and the phase angle of the vector represents the acrophase (ϕ, ZT, Zeitgeber Time). The ellipse at the tip of the vector represents the confidence limits for the amplitude-acrophase pair. SC: Standard conditions, RF: random feeding, 24D: total darkness, ML: fish fed at mid-light, MD: fish fed at midnight.

## Data Availability

Data included in the article.

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
