# Peer review of "Daily Rhythms in the IGF-1 System in the Liver of Goldfish and Their Synchronization to Light/Dark Cycle and Feeding Time"

_animals, 2022, doi:10.3390/ani12233371_

Round 1

Reviewer 1 Report

General comments

This work describes in detail the effects of the nutritional status and light/dark cycle on the expression levels of various components of the IGF-1 system in goldfish.  The manuscript is well-structured materials and methods are adequate and results are well supported by experimental evidence.

Gene nomenclature should be checked and corrected throughout the manuscript. The general rule is that gene symbols are italicized and protein symbols are not italicized, however, there are variations among organisms. More specifically for fish Gene symbols are italicized, with all letters in lowercase.

Authors should acknowledge the in the introduction the existence of additional IGFBP in goldfish namely IGFBP3, 5 and 6 although these paralogues were not the focus of this paper.

Minor comments

Suggestion: A schematic image of the IGF1 system would help the reader follow the text and description of the function of each component in the introduction.

Abstract:

L27: Missing punctuation Full Stop, after the word goldfish. Also, I suggest that authors start the next sentence with “In addition/Also” instead of “Besides”.

Introduction:

L46-48: This last sentence is hard to follow authors should reword it to clarify.

L66: Missing a reference citation after “…tyrosine kinase domains”.

L72: Correct Cyprinids “4Rc”.

L76 & L79: Missing references/citations after “… growth factor binding proteins (IGFBPs).” And “… local IGF-1signalling.”

Authors should clarify if the described characteristics of IGF-1 are observed in teleost species or in other vertebrates.

L87: Clarify which rounds of whole genome duplication are involved in the IGFBP duplications.  Is this ancestral chordate genome duplications? And/or additional lineage specific duplications.

Which lineages experienced gene loss?

L89: Correct typo” …early ancestral gen IGFBP”, also gene symbol should be italicized.

L93: Gene symbol italicized.

L97: Correct IGF-I to IGF-1, keep nomenclature consistent throughout the manuscript.

L104: Correct IGF-I to IGF-1.

L110-113: This sentence is confusing, authors should reword it to clarify their point.

L120: Suggestion start the sentence with “The”.

Materials and Methods

L137: Missing information on the RNA extraction procedure and cDNA synthesis.

Results

Figure 1 legend: What is igf1rs?

L269-270: Authors should acknowledge the existence of additional IGFBP in goldfish namely IGFBP3, 5, and 6, and mention that these paralogous were not investigated in this manuscript.

Discussion

L388: igfr1a expression levels were not similar values in all studies tissues, igfr1a expression level was significantly higher in Gill and Heart. The authors should adjust this sentence.

Similarly, to igfr1b authors should also present a comparative overview of igfr1a in other species.

L406: Gene symbol missing an F and should be italicized IGFBP.

L447: Specify which clock genes.

L534-536: This sentence is confusing and should be revised. What do authors mean by IGF concentrations? Also correct Fail= failure 

L539: Revise English in the last paragraph of the discussion.

Conclusion

L549: We demonstrate (singular) correct typo.

L552: Rephrase/reword as this sentence does not make sense “emphasizing the relevance of distinguish”.

Reviewer 2 Report

1. Original Submission

1.1. Recommendation

Minor Revision

2. Comments to Author:

Ms. Ref. No.: animals-2057123

Title: Daily rhythms in the IGF-1 system in the liver of goldfish and 2 their synchronization to light/dark cycle and feeding time

Overview and general recommendation:

The study is opportune as there is scarce information about the relevance of insulin-like growth factor-1 and its binding proteins as potential rhythmic outputs of the liver clock in fish. Present data provide relevant information about the IGF-1 system checked in a wide variety of tissues and the results support differential tissue distribution of different subtypes of IGF-1 receptors. The results in gills and gonads can help to understand the osmoregulatory and reproductive functions of the IGF-1 in fish together with the hepatic release, that evidences an endocrine role.

2.1. Major comments:

The authors offer scarce information about the distribution of the animals in the tanks along the experimental period. The M&M explain that the fish have been kept in 60 liters’ tank that are small for the amount of fish involves on the study.   

2.2. Minor comments:

Line 125: The initial number of fish should have been of more than 200 fish. Please stated the initial amount and how this fishes have been maintained as they have need bigger or large amount of specific tanks that are not adequately explained.  

Line 160: Kindly explain the aim of change of the experimental organization from 42 to 36 and the change in the cycle sampling.

Line 165:  different cycle arrangement from 5 to 2, may be it has been 5 to 21? Why this time has been different? Working with fish, mortality is normal and extra animals must be initially added in order to prevent mortality events.  

Reviewer 3 Report

This study by Alonso-Gómez et al. explores the RNA fluctuations over time of insulin-like growth factor-1 and its binding proteins and receptors in goldfish exposed to different light/darkness and feeding regimens.

The authors start out by providing an RNA expression overview of igf-1, igf-1 receptors (a and b) and igf-1 binding proteins (1 and 2 a-a, a-b, b-a, b-b) in several different tissues of the goldfish. They show that the liver - amongst the tested tissues- is the one with the highest igf-1 expression. Therefore, it is the organ that they will use for their further studies. They continue setting up three different conditions for three different groups of goldfish, measuring the expression level of the previous mentioned genes at 6 different time points (every 4 hours starting 1 hour after the feeding). They show that gene expression over time is affected by changes in the light/darkness and feeding conditions - significantly for igf-1 and igf-1 binding proteins - while not significantly for the receptors. In the last experiment they compared two groups having both the same light/darkness cycle of 12L:12D but with the first group fed once per day in the middle of the day and the second group fed once per day in the middle of the night. The gene expression oscillation phase is shifted between the two groups according to the feeding, indicating that it has a major role as a zeitgeber, affecting the rhythmicity of IGF.

Finally, they provide a resuming visual overview of the effects of light/darkness and feeding on the acrophase (point of maximum expression of an oscillating gene) and amplitude of the fluctuation for igf-1 igf1bp1a-b and igf1bp1b-b. This comparison offers - at a glance - how food and day rhythm manipulations affect the different genes in different conditions. Although this study offers a comprehensive overview about the daily rhythmicity of IGF-1 and IGF-1 related genes expression in an organism exposed to different zeitgebers, there are few major questions that need to be addressed to not result in speculative conclusions. 

Mayor Points

Point 1:

While the reviewer fully understands the extra challenges posed to authors whose native language is not English, the help of a native speaker in reviewing and editing the manuscript prior to submission is essential. Here, the reviewer would like to give only a few representative examples to fortify this assessment: 

Simple Summary: 1st line: “is a peptide that act both” – singular: acts

Line 27: ‘’goldfish, Besides’’. Correct the comma or the capital letter

Line 31: ‘’persist’’. ‘’persists’’

Line 44: “, where induces” ? where it induces?

Line 45: ‘’exert’’. ‘’exerts’’

Line 52: ‘’in most species so far studied’’. Better ‘’in most of the species studied so far’’

Line 59: ‘’as’’. ‘’such as’’

Line 59-60: remove all the “the”

Line 95: remove “of the IGFBPs” 

Line 99: ‘’viewed in depth’’. Better ‘’explored thoroughly’’

Line 139: “The mRNAs expression of the igf-1, igf1ra and” – might want to change to - smRNA expression of igf1, “

Line 140: ‘’in those locations’’. Remove (it is clear already) or change with ‘’in those tissues’’

Line 149: ‘’conducted for suppression the light-dark cycle zeitgeber’’. For ‘’suppressing the’’ or ‘’for suppression of the’’

Line 160: ‘’stablished’’. ‘’established’’

Point 2:

Line 150, 163, 232: One of the major points presented in the study is how different food delivery alters IGF-1 and IGF-1 related gene expression. To do so the authors often employ nocturnal feeding, which is poorly described in material and methods.

It is indeed not indicated whether the fish are fed in the darkness or if the lights are briefly turned on during the feeding period. This would ensure the feeding but on the other hand destroy the circadian rhythm crucially because of the light source and not because of the food source.

In the opposite scenario - when lights are never turned on during the feeding period - how do the authors check that fish actually eat at that specific time?

Fish belonging to the group 24D – darkness all the time - could just eat randomly from time to time and the effects on their gene expression oscillation would not be derived just by the light zeitgeber but from a combination of no light + random food intake. Supporting this conclusion is the fact that the gene oscillations of group 24D are more similar - at a first glance - to the RF group (Figure 2 B-C, Figure 4 B-C).

Therefore, a proof of eating in darkness is required or the statement about the 24D group should assume that this group both wakes/sleeps randomly and eats randomly in order to avoid being speculative.

Line 300: The 12 hours shifted feeding presents a similar concern. How can the authors demonstrate that the MD group actually eats during the night at the precise ZT18 time point? Without any proof one could assume that those fish could eat at the earliest light time point when they wake up, which would result in a ZT24 feeding with only a 6 hours shift.

This could be supported by Figure 5 A-B, where the highest values (H5 and H13) have a shift of 8 hours (closer to a +6 than to a +12 shift). Figure 6 C-D lowest points are 21H vs 13H, again a shift of 8h. Figure 6 G-H highest points are 17H vs 9H, a 8 hours shift.  

Another important point is that a night feeding regimen, if eaten, will also destroy the wake/sleep schedule of those fish compared to the controls and the effects shown would be therefore brought not only by food as a zeitgeber but also by a wake/sleep cycle nocturnal destruction/disturbance.  

Minor Points

Point 1:

Material and methods are quite comprehensive - but some major information is missing concerning the specimen used. The number of fish housed per tank should be declared as well as their sex ratio. It is not mentioned anywhere if experiments have been performed in male or female or in both sexes. Also it could be useful to give an estimation of the age of the fish, even if those were taken directly from a supplier.

Point 2:

Line 213. The authors indicate to have characterized the expression of different oscillating genes in different tissue, claiming that (taking as example igf-1) the most abundant expression was found in liver. Due to the fact that those genes are supposed to oscillate (that is the core of the whole study), it is important to provide a time point relative to when that analysis was performed, and state that at that time point, at that age in that fish the expression resulted as shown. It is indeed possible that at another time point, due to fluctuations, genes would show higher expression in other tissues. One would want to refrain from this kind of speculation. 

Point 3:

Line 493:  how can the authors tell - if it is 4 hours prior the next feeding or 20 hours after the last feeding - if you feed once a day? I suggest to rephrase or to do another experiment with an additional feeding or a starving period to check if the fluctuations are triggered by the anticipation of the feeding procedure (therefore is 4 hours prior), or by a leftover effect due to the last feeding (therefore they won’t forecast anything, just have the residual effect from the last feeding).

Point 4:

Line 547: “Our results support that IGF-1 from the liver is released into blood evidencing an endocrine role of hepatic IGF-1, while the broad local production of this peptide may exert paracrine/autocrine actions in multiple organs in goldfish”. I suggest to rephrase or delete this sentence because it is very speculative. The authors never conducted an analysis on blood and only have (partial) indirect proofs for such a statement.
